# Modeling and Processing of Smart Point Clouds of Cultural Relics with Complex Geometries

**Su Yang** [1], **Miaole Hou** [2,*], **Ahmed Shaker** [3] **and Songnian Li** [3]

1　College of Geosciences and Surveying Engineering, China University of Mining and Technology, Beijing 100083, China; yangsu@student.cumtb.edu.cn
2　School of Geomatics and Urban Information, Beijing University of Civil Engineering and Architecture, Beijing 100044, China
3　Department of Civil Engineering, Ryerson University, 350 Victoria St., Toronto, ON M5B 2K3, Canada; ahmed.shaker@ryerson.ca (A.S.); snli@ryerson.ca (S.L.)
*　Correspondence: houmiaole@bucea.edu.cn

**Abstract:** The digital documentation of cultural relics plays an important role in archiving, protection, and management. In the field of cultural heritage, three-dimensional (3D) point cloud data is effective at expressing complex geometric structures and geometric details on the surface of cultural relics, but lacks semantic information. To elaborate the geometric information of cultural relics and add meaningful semantic information, we propose a modeling and processing method of smart point clouds of cultural relics with complex geometries. An information modeling framework for complex geometric cultural relics was designed based on the concept of smart point clouds, in which 3D point cloud data are organized through the time dimension and different spatial scales indicating different geometric details. The proposed model allows smart point clouds or a subset to be linked with semantic information or related documents. As such, this novel information modeling framework can be used to describe rich semantic information and high-level details of geometry. The proposed information model not only expresses the complex geometric structure of the cultural relics and the geometric details on the surface, but also has rich semantic information, and can even be associated with documents. A case study of the Dazu Thousand-Hand Bodhisattva Statue, which is characterized by a variety of complex geometries, reveals that our proposed framework is capable of modeling and processing the statue with excellent applicability and expansibility. This work provides insights into the sustainable development of cultural heritage protection globally.

**Keywords:** cultural heritage; point cloud; 3D model; information modeling; complex geometry

## 1. Introduction

Digital documentation of the status of cultural relics is essential for their protection and scientific studies during the restoration and renovation process [1]. Associating the abundant semantic information of cultural relics with the visualized three-dimensional (3D) models poses certain challenges for the documentation process. The Strategic Action Plan for the Implementation of the World Heritage Convention 2012–2022 noted that "The Outstanding Universal Value (OUV) of World Heritage sites is maintained" and "Heritage protection and conservation considers present and future environmental, societal and economic needs" [2]. Heritage values refer to the meanings and values that individuals or groups of people bestow on heritage, including historical, aesthetic, economic, social, and scientific values [3]. Existing information models that lay a solid foundation for maintaining cultural relic values are mainly based on parametric modeling methods, including 3D GIS, Building Information Modeling (BIM), and Heritage Building Information Modeling (HBIM). These approaches are essentially derived from 3D modeling technology, and support cultural heritage management, protection, monitoring, analysis, and research [4–6].

However, 3D models only provide visual effects and visual analysis, but lack semantic information and knowledge related to the cultural relics [7–10].

Cultural relics, such as the Bayon Temple [11] and the Digital Michelangelo Project [12], usually have complex geometric structures and highly detailed geometric textures. The parametric modeling method, which is a popular method for 3D modeling, is not suitable for constructing 3D models of culture relics, because it requires a large amount of manual work and oversimplification to model complex geometries of culture relics [13]. Another popular 3D modeling method, named 3D GIS, is mainly used for visual display and spatial analysis of macro scenes. BIM and HBIM have been developed for 3D modeling of buildings with cultural heritage. Despite the complex geometric features that are necessary for cultural relic modeling, the damaged parts on the surface of cultural relics, such as traces of weathering, are likely to cause subtle geometric deformation, which plays an important role in cultural relic protection. Thus, with respect to the 3D digitization of cultural relics, the level of representation of geometric details is critical. Moreover, adding semantic information and knowledge to the 3D models of cultural relics can significantly expand the application range of these information models and improve the level of information acquisition for those who use the models. As such, developing an information modeling method with more comprehensive applicability is necessary to express high-level geometric details of cultural relics, and to be associated with semantic information.

Smart point clouds are a new data platform that addresses the situation in which a large amount of discrete spatial information from active remote sensing technology is underutilized, which hinders data mining [14]. However, this is a general concept without considering the characteristics of the cultural heritage field. Focusing on cultural heritage, Poux et al. [15] adopted the concept of smart point clouds and developed a built heritage information system prototype based on high-resolution 3D point cloud data. This study emphasized the management and storage of geometric information and semantic information for cultural heritage, but placed little focus on the fusion of geometric scenes and the diverse information of cultural relics. Cultural relics involve multispatial scales, multitemporal stamps, and multidisciplinary scenarios. Researchers have mainly conducted cultural relic-related studies in macro scenes where they are located. In addition, cultural relics undergo deformation over time, which is usually caused by damage or natural disasters, and needs to be recorded from the temporal perspective. Multidisciplinary integration refers to organizing the information that stakeholders are concerned about from different perspectives into a 3D scene to facilitate information mining. In this manner, the relationship among the multidimensional information of cultural relics can be established in the context of space, time, and multiple disciplines.

In this work, a novel information modeling method was proposed based on 3D point cloud data to meet the above-mentioned demands through modeling complex geometric structures with detailed geometric textures of cultural relics, and linking them with semantic information and knowledge. This model presents a new solution for modeling multidimensional information of cultural relics by integrating a high-precision visualization 3D model with multidisciplinary semantics. This model is not limited to a specific type of cultural relic, but serves as an open and extensible model framework that associates point sets in the point cloud data with valuable information, forming a spatiotemporal and semantic information platform. Moreover, this 3D point cloud data-based model is able to update and enrich the digital model of cultural relics in the context of dynamic management of space, time, semantics, files of cultural relics, and their specific environment. To briefly summarize, our proposed smart point cloud framework based on high-density (in the x, y, z direction) multispatial scale 3D point cloud datasets is able to:

- Provide a variety of information and support researchers from different disciplines to study cultural relics; for example, any semantic information, or associated files, can be attached to the point cloud.

- Conduct virtual restoration of damaged cultural relics. Based on the acquired point cloud data of a damaged artifact, it is feasible to reconstruct its original geometries using a computer.
- Investigate spatial distribution of damaged information of cultural relics.
- Measure geometric deformation of cultural relics varying with time.
- Provide a conceptual model for developing a cultural relic information management system. Based on the conceptual model, the logical data model and physical model are established based on the selected objects.

This paper is organized as follows. Section 2 outlines the existing attempts to apply point cloud data and current 3D modeling methods to modeling cultural relics, including their shortcomings. Section 3 presents the basic framework of our proposed information model based on 3D point cloud data and provides details about each step, analyzing its main components and its purpose. In Section 4, a famous cultural relic, the Dazu Thousand-Hand Bodhisattva Statue, which is part of the world heritage site of Dazu Rock Carving located in Chongqing, China, was selected as a case study to illustrate the performance of our proposed model. Finally, the results are discussed in Section 5, followed by conclusions in Section 6.

## 2. Related Work

### 2.1. 3D Point Cloud

The point cloud model is able to express the geometric details of cultural relics. Three-dimensional point cloud data, as the main source for constructing 3D models with complex geometries in the cultural heritage field, are composed of a massive number of spatial points, i.e., (x, y, z), to represent geometric shapes [16,17]. The 3D point cloud has been widely used for digitalizing archives [18], physical reproduction of artworks with high fidelity [19], geometric computing and analysis [20], online demonstration of cultural heritage using remote control [21], and virtual restoration [22] and monitoring of cultural heritage [23]. Therefore, the point cloud is an essential type of primary data to aid cultural heritage management and protection.

Unmanned aerial vehicle (UAV) photogrammetry and 3D laser scanning technologies have been widely applied for the 3D geometric reconstruction of cultural heritage with image textures [24,25]. Moreover, these two techniques can be integrated to generate 3D point cloud data at different spatial scales [26]. Specifically, UAV photogrammetry technology generates dense point clouds representing the surface of cultural relics based on the prior parameters concerning exterior orientation and camera calibration. The 3D models of cultural relics can be further digitally constructed using either automatic dense image matching techniques or interactive methods by extracting man-made features and vector information from point clouds [27]. As a result, 3D scene reconstruction and visualization are realized by point cloud interpolation, simplification, and texture mapping. UAV photogrammetry is mainly adopted for 3D reconstruction of large scenes, such as cultural heritage protection areas, ruins, and building groups.

In the past 20 years, 3D laser scanning technology has provided technical methods for the high-precision digitization of cultural relics. Three-dimensional laser scanning technology outperforms other traditional technology because it is able to detect complex geometric shapes with high accuracy to provide more detailed geometric information [9]. This technology typically relies on terrestrial laser scanning (TLS) and hand-held laser scanning to directly acquire 3D spatial information of cultural relics, which corresponds to a dense point cloud dataset composed of 3D spatial coordinates [28,29]. In this manner, 3D models of cultural relics are generated through resampling, denoising, and cleaning of the obtained dense point cloud data. Because the increase in the distance between lidar sensors and the measured cultural relic leads to the decrease in the scanning accuracy, 3D laser scanning technology is mainly applied to 3D reconstruction of cultural relics in medium and small scenes, e.g., small cultural heritage protection areas, independent buildings, and sculptures. Furthermore, based on the 3D information, the color of each point can

be provided by sensors through an additional passive channel. This information allows real-time rendering of 3D models [30]. In addition, 3D laser scanners of sensors are used to obtain the laser reflection intensity, which assists point cloud segmentation and object recognition [31]. These additional parameters, including reflection intensity and color, enrich the point cloud attributes, improving the object representation in 3D scenes. [32,33]. However, such additional information places little significance on semantic modeling of cultural relics.

The majority of point cloud-based models are mainly applied to visualization and visual analysis, and are unsuitable for most other spatial analyses due to a lack of semantics. Point clouds suffer from several structural limitations, resulting in indirect exploitation via human-interpreted deliverables [14]. In this process of data processing, it is likely the geometric details, and the advantage of the high accuracy of the point cloud data, will be lost. Using laser scanners or image-based reconstruction methods to generate a point cloud dataset, users must first identify collections of points that belong to individual surfaces, and fit surfaces and solid geometry objects appropriate for the analysis [34]. Three-dimensional point cloud datasets, with geometric, radiometric, and semantic properties—i.e., the rich point cloud paradigm—based on dense and semantic 3D point clouds, such as in the case of city 3D models, are only suitable for micro to mesoscale models, and not large spatial scale 3D modeling [35]. In cultural relics with intricate geometric details and shapes, the concepts of the rich point cloud or the smart point cloud can be used to address the problem of the lack of semantic information in 3D point cloud datasets.

### 2.2. BIM and HBIM

In cultural heritage, BIM and HBIM are derived from point cloud data through parametric modeling. The management of cultural heritage is realized through additional semantic information. BIM is a standardized modeling process applied to architecture that allows the creation of 3D building models and their expression in conjunction with digital data, text, images, and other types of information [36,37]. BIM has conferred numerous benefits to the construction field, such as enhanced design visualization, improved data exchange, improved productivity, and excellent product quality. These characteristics have led to the gradual application of BIM technology in historic building conservation [38]. Moreover, BIM models can be used to carry out structural simulations, stress analyses, and protection analyses [39]. It is thought by specialists of point cloud processing that this knowledge helps improve the automation, accuracy, and resulting quality of modeling [40].

Via geometric models, HBIM can be applied to the comprehensive protection and restoration of historical buildings, the interpretation of their detailed history, and the recording of various information relating to architectural heritage preservation and restoration status over time [41]. Integration of geometric data and semantic information is essential for managing the architectural heritage, either for its restoration, its maintenance, or for the dissemination of the richness entailed in these historical assets [42]. By extending the capability of the BIM platform, HBIM geometric elements and semantic ontology knowledge are joined in a unified BIM environment, including accurate parametric modeling using computer graphics, automatic semantic segmentation of 3D point clouds from reality-based modeling, spatial information management, analysis by GIS, and knowledge modeling by ontology [43,44]. However, BIM is generally based on the scan-to-BIM process, which allows the generation of 3D models from point clouds. To transform from a point cloud to BIM or HBIM, the reverse modeling process causes accuracy problems, such as in the case of the mesh quality, depending on the point cloud density [45–47]. Significantly, the difficulties increase because of the peculiarity of the structural behavior and the singularity of the geometrical shape. Thus, it is challenging to balance the geometric accuracy of 3D models with parametric modeling [48]. Therefore, there are two limitations in the application of BIM or HBIM to cultural relics. The first is that BIM and HBIM are widely used in buildings or historic buildings, but are not applied to other artifacts. Second, it is

difficult to model the complex structure of cultural relics based on the parametric modeling method because the loss of surface details is unavoidable in the process of 3D modeling.

Thus, the information modeling of cultural relics with complex geometries involves the trade-off between geometric details and rich semantic information or knowledge. The point cloud data can be more readily used for expressing the high-precision geometrical information of cultural relics but lacks semantic information. However, point cloud data provides significantly less semantic information or knowledge than BIM and HBIM. Parametric modeling methods, such as BIM or HBIM, cannot guarantee high-precision geometric information, but involve semantic information. Therefore, the contribution of this paper is to propose a new information modeling framework based on 3D point cloud data that balances the maintenance of geometric details and engagement of semantic information.

## 3. A General Framework for Smart Point Clouds of Cultural Relics

Figure 1 shows the overall modeling process based on the proposed smart point cloud framework for 3D modeling of cultural relics with complex geometries.

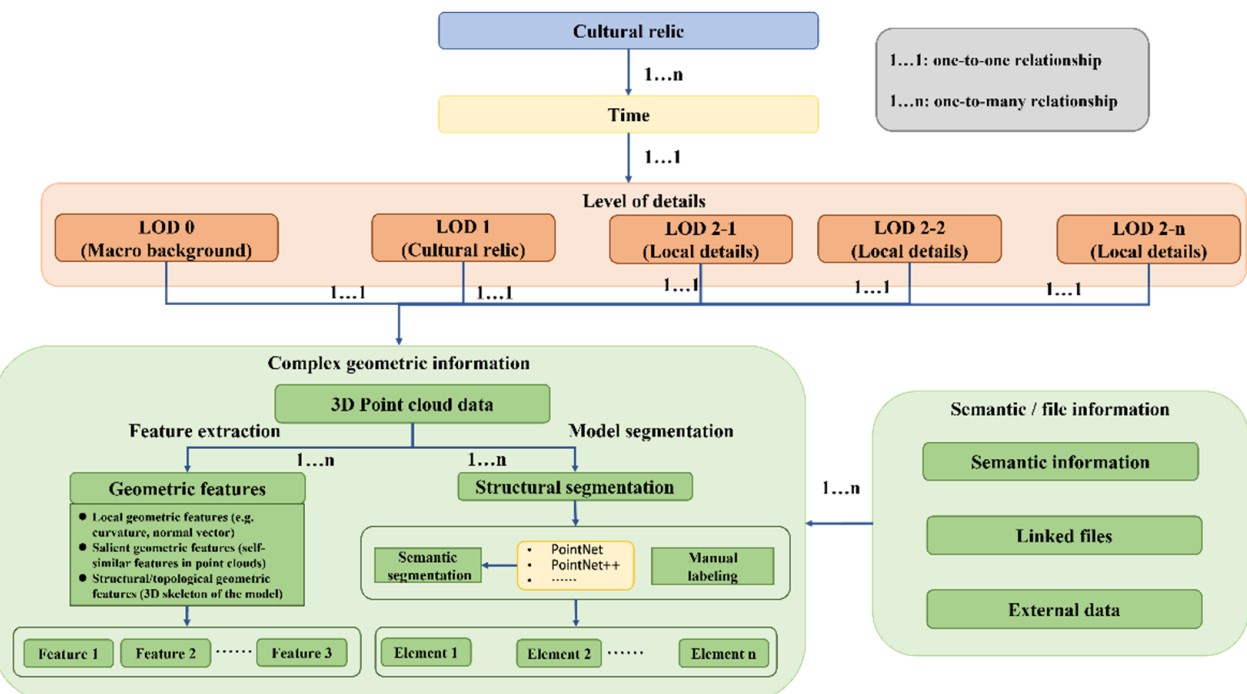

**Figure 1.** Framework of a smart point cloud of cultural relics with complex geometries.

### 3.1. Time

Time information is a necessary indicator for tracking the geometric deformation of cultural relics. Geometric deformation is usually caused by changes in the microenvironment, such as cracks caused by temperature and humidity, tilting of the geometric structure caused by uneven pressure, natural disasters, and other factors. A better understanding of the causes of geometric deformation enables evaluation of the safety and appropriateness of cultural heritage management, and heritage protection can support auxiliary decisions for the sustainable conservation of cultural relics. In addition to the timestamp indicating when the 3D point cloud data of a cultural relic is collected (i.e., $T_{col}$) to investigate whether it has been deformed, the time information regarding the same cultural relic also includes the timestamp after natural disasters such as earthquakes, floods, and rainstorms (i.e., $T_{nd}$), and the timestamp after protective restoration (i.e., $T_{pr}$). In this manner, the time information with regard to a cultural relic is composed of a time series in the proposed information model, which can be described as $T_{CR} = (T_{col}, T_{nd}, T_{pr})$. Therefore, the time

information of 3D point cloud data is first provided in the information model framework, and acts as a basis to organize other information.

## 3.2. Levels of Detail (LOD)

The LOD of 3D point cloud data are analyzed from the spatial perspective. The hierarchical framework indicating the multispatial scale of cultural relics, including the macro-, meso-, and micro-scenes of the object, involves a hierarchical relationship under different observation scales. Therefore, a hierarchical framework with different spatial scales is required to connect the acquired data to fulfill the requirements of research from different disciplines. Multispatial scales mean that 3D point cloud data of multiple scales are required to meet various cultural relics modeling demands [49,50]. It is necessary to select data acquisition methods with different spatial resolutions according to the specific application requirements. The data volume and point cloud accuracy of the model meet the needs of current research. The hierarchical LOD are divided into three levels in this framework as follows.

LOD0: macro-scene, i.e., the surrounding environment where the cultural relics are located.

LOD1: meso-scene, i.e., the main research object.

LOD2: micro-scene, i.e., part of the research object or details on cultural relics.

## 3.3. Complex Geometric Information

### 3.3.1. Feature Extraction

Geometric feature extraction refers to extracting the representative geometric information contained in the 3D point cloud data of cultural relics [51–54]. Some features are obvious and easy to extract, such as those obtained by projection or measurement, whereas other geometric features must be extracted by running complex algorithms. For example, as shown in Figure 2a, by calculating the degree to which the local curvature of the neighborhood point cloud changes, the roughness of the cultural relic's surface can be obtained, which can be used to identify its sharp features. In particular, for certain microscopic features, geometric descriptors need to be expressed, such as the roughness of the model surface, and the extraction of convex and concave parts, which are geometric features used to identify and splice cultural relic fragments. Local surface descriptors representing the geometry of local regions of the surface support extraction of salient geometric features, which assists the search for self-similarity among 3D models [55]. The purpose of local surface descriptors is to automatically identify salient features that have a multitude of similar occurrences across the given surface. The skeleton line usually holds essential information that supports the stability of the object. The Laplacian operator is a feasible method to extract the skeleton line from the point cloud and mesh. This operator iteratively and smoothly contracts the point cloud or mesh to form a point set that provides the shape of a skeleton. Accordingly, a number of key points are selected to generate a fitted skeleton by simplifying the point set [56]. Figure 2b shows the result of the skeleton lines as a basic geometric shape of an object and the topological relationship between the fingers. The skeleton line extraction is carried out using the method proposed by Au et al. [57], thus illustrating the performance of this applied method for extracting the skeleton line of cultural relics.

An object is made up of some essential elements that correspond to technical terms in cultural heritage. The element is the cognition of the structure of cultural heritage. Geometric structure (i.e., different regions with semantics in the point cloud and the topological relationship between those regions) provides significant assistance in the analysis of the geometric form and structure of cultural heritage objects. For example, extracting repeated geometric textures of the same shape or extracting the same type of damaged information leads to similar geometric deformation. From another perspective, point cloud segmentation can be understood as adding one or more attributes to the spatial point, categorizing points and point sets, and assigning additional information to them.

The result can be expressed with meaningful semantics (e.g., words) in line with people's understanding of the geometric structure of the cultural heritage object.

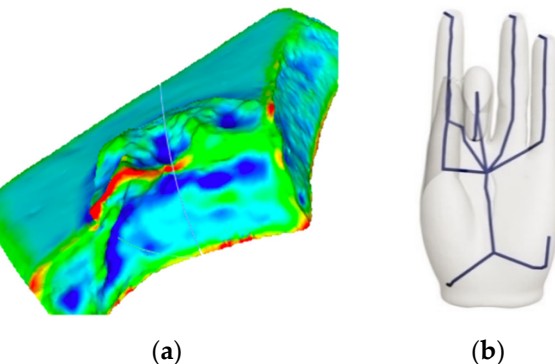

(**a**)                                                          (**b**)

**Figure 2.** (**a**) The sharp features of a cultural relic fragment. (**b**) The skeleton line can show the basic geometry of the hand and the topological relationship between the fingers.

Point cloud segmentation aims to correctly partition the scene and essential elements of cultural relics. These partitioned elements aid the cognition of the geometric structure of cultural relics. To achieve this goal, the automatic and accurate partition of the geometric elements of cultural relics poses certain challenges, which usually involve point cloud segmentation and clustering algorithms [58–61]. However, in practice, bottlenecks exist regarding the segmentation of 3D point cloud data of cultural relics, which restricts the automation of the 3D point cloud-based framework proposed in this paper. The prominent bottlenecks mainly relate to two aspects. First, the clustering and segmentation of 3D point cloud data used by artificial intelligence methods are mainly applied to 3D city modeling, and acquiring training datasets for cultural relics is difficult. Second, cultural relics are characterized by special features, which cannot be easily addressed by the existing methods. As a result of the above-mentioned bottlenecks, point cloud segmentation mainly relies on manual annotation. Currently, commercial software is generally used for manual segmentation and semantic annotation of segmented regions. Nonetheless, successful attempts have been made that demonstrate the potential for practical application. For example, following an initial partition of point cloud data, a RANSAC-based plane fitting algorithm has been used to segment point clouds into meaningful subsets [62]. E. Grilli et al. [63] analyzed the efficiency of the capability of geometric covariance features supporting cultural relic point cloud segmentation and classification.

### 3.3.2. Semantic Linking

Semantic linking is the process of linking the 3D point cloud data, as a whole or a subset (i.e., an element), with semantic information and related files. Semantic information mainly includes terminology, semantic descriptions, spatial location, geometric characteristics, damage investigation results, heritage values, and other relevant information. The formats used to explain cultural relics include audio, video, text, image files, and 3D models.

3D point cloud data has spatial properties (i.e., (x, y, z)), color properties (i.e., (x, y, z, R, G, B)), and reflection intensity properties (i.e., (x, y, z, R, G, B, ref)). Structured data is formed by storing point cloud data in a database (e.g., Oracle or MySQL) according to the defined data structure. In addition to spatial coordinates organized as (x, y, z), which are necessary properties of point cloud data, other semantic information, such as descriptions of damage investigation results, heritage value is organized in terms of an individual point or a point cluster. These additional semantic properties are usually generated by clustering and segmenting 3D point cloud data of cultural relics. In addition, these properties are associated with specific files by setting the property of a unique identifier or the path to a file (e.g., a unique URL). Thus, semantic properties of cultural relics are related to files in

differentiated ways, which distinguish one from another. Our proposed smart point cloud framework essentially reflects the relationship between individual points or point clusters and other multidimensional information, which together constitute a 3D point data-based information model. It should be noted that the framework is also able to express complex and interdisciplinary content. Therefore, there is a possibility of an intersection among different 3D point datasets; that is, a point included in several point cloud datasets is likely to be assigned with multiple attributes or elements.

## 4. Case Study

We conducted a case study using the Dazu Thousand-Hand Bodhisattva Statue in Dazu Rock Carvings, Chongqing, China, which is a representative cultural relic with complex geometries, to demonstrate the performance of our proposed smart point cloud framework.

### 4.1. Cultural Relic Background

The Dazu Rock Carvings are a famous world cultural heritage site located in Dazu district, Chongqing city, China. As shown in Figure 3, the Dazu Thousand-Hand Bodhisattva Statue is a gilded clay and cliff stone sculpture with thousands of hands. The diverse hand postures and the held objects indicate different Buddhist meanings, generating a huge number of complex geometries. The statue is 7.7 m in height and 10.9 m in width. The rock face is carved with 830 Guanyin hands gripping different accessories. The surface of the stone-carved statue is gilded, and the objects are painted in various colors. Due to the statue's age, the gold leaf and colored paintings on the surface of the statue have fallen off, exposing the rock mass. The fingers of some Guanyin hands are broken and missing.

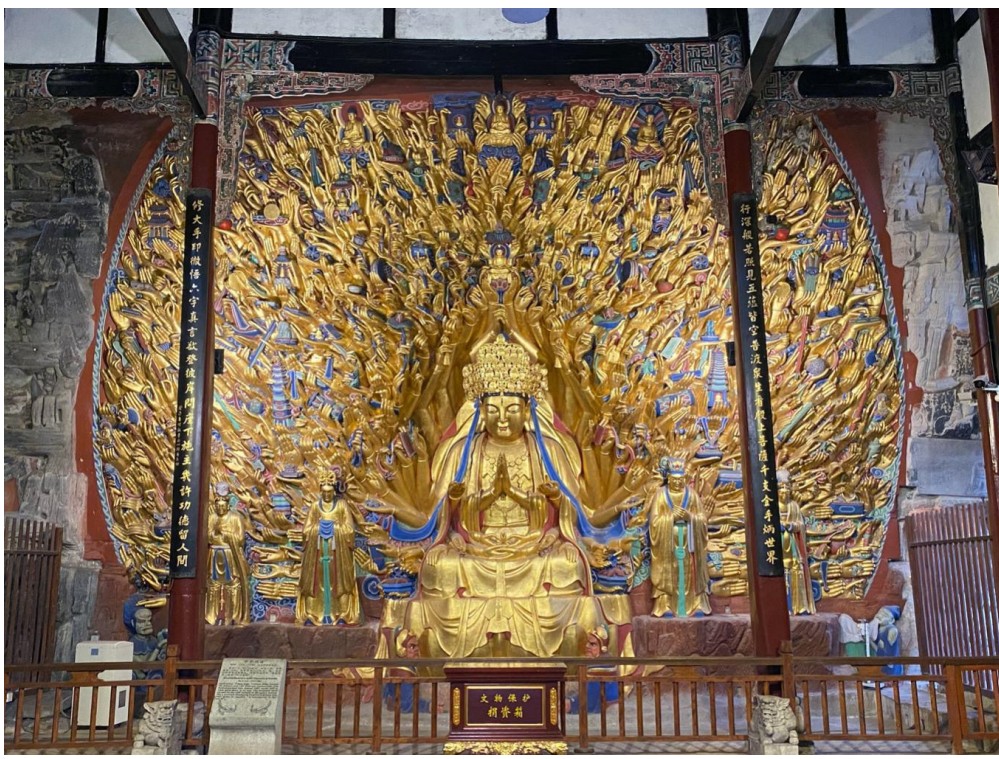

**Figure 3.** The Dazu Thousand-Hand Bodhisattva Statue located in Chongqing, China (2020)—photo by Su Yang.

### 4.2. Smart Point Cloud of Dazu Thousand-Hand Bodhisattva Statue

Based on the conceptual smart point cloud framework that was introduced in Section 3, the Dazu Thousand-Hand Bodhisattva Statue was used to illustrate how this model is applied to cultural relics with complex geometries. As shown in Figure 4, the extended framework regarding the smart point cloud includes the time at which the data was

collected, the scenes of three spatial scales, the point cloud data collected for different levels of detail (i.e., LOD0, LOD1, LOD2), and the related semantic information and associated files.

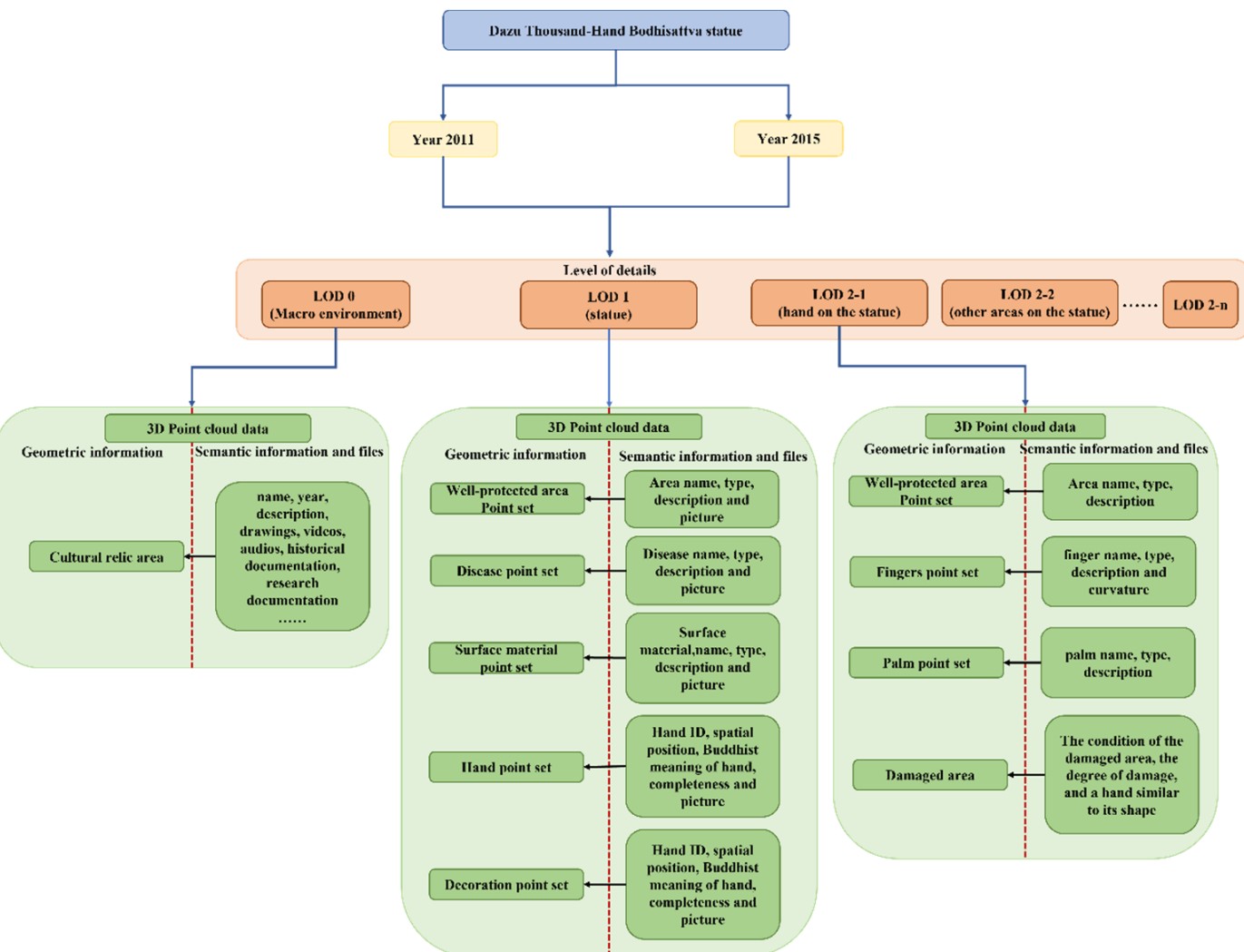

**Figure 4.** The framework of the smart point cloud for The Dazu Thousand-Hand Bodhisattva Statue.

Three-dimensional point cloud data were collected for the Dazu Thousand-Hand Bodhisattva Statue in 2011 and 2015. During 2011 and 2015, a restoration project was undertaken for the statute. The data were collected according to the three detail levels that were introduced in Section 3.2, namely LOD0, LOD1, and LOD2. It is noted that LOD0 data was not collected in 2011. Table 1 summarizes the technology and equipment used for the 3D point cloud data collection. More details are provided below:

- LOD0 in the macro-scene was generated from the 3D point cloud data that were collected using UAV photogrammetry technology in a large area where the cultural relics are located (see Figure 5). A fixed-wing UAV system coupled with an RTK and equipped with a Sony A7RII full-frame camera was used. The area was around one square kilometer. The UAV's flying height was about 100 m and more than 8000 images were produced. The computed Root Mean Square Error (RMSE) of seven targets was 2, 2, and 3 cm in the X, Y, and Z direction, respectively. These parameter values meet archaeological and research requirements. The collected 3D point cloud data were linked to semantic information, including basic information such as name, age, background, and description, in addition to the information extracted from

the collected historical documents, research papers, audiovisual materials, project drawings, etc.

- LOD1 in the meso-scene was generated from the 3D point cloud data of the Dazu Thousand-Hand Bodhisattva Statue that were collected using TLS technology (see Figure 6). This scene recorded the statue with a total of 50 million points. The data collection area was around 12 m long and 8 m high. This area was bigger than the statue because we took the edge area of the statue into account. Then, a variety of damaged areas, different surface materials (e.g., golden, painted, exposed rock), all of the Guanyin hands, and decoration objects held by the hands, were extracted, during which a unique identification was assigned to each. For instance, the damaged area was associated with semantic information, such as the type and extent of damage. After applying the segmentation or clustering, the area of the damaged parts was calculated from the point clouds, which contain the semantic information (e.g., damage type) attached to each point or point patch.

- LOD2 in the micro-scene was generated from the 3D point cloud data that were collected using handheld 3D laser scanners. Because the point cloud data collected in this manner were dense (e.g., sample spacing of 0.1 mm), the subtle geometric forms on the surface of the statue were able to be clearly observed (see Figure 7). Even the slightly damaged areas that did not result in geometric deformation can also be marked by our model. The point set representing a finger was linked with semantic information describing its preservation status. As shown in Figure 4, LOD2 can be composed of multiple 3D point cloud datasets (i.e., LOD2-1, LOD2-2, . . . , LOD2-n). In spite of the damaged areas, LOD2 also includes other areas, such as the well-preserved areas of cultural relics, of which the expression form is consistent with the description method of the hand on the statue, i.e., LOD2-1. In the scene of LOD2, point cloud data for a total of 829 Guanyin hands were collected, comprising 320GB in total. In addition, about 4000 photos were taken to record the status of each Guanyin hand under visible light.

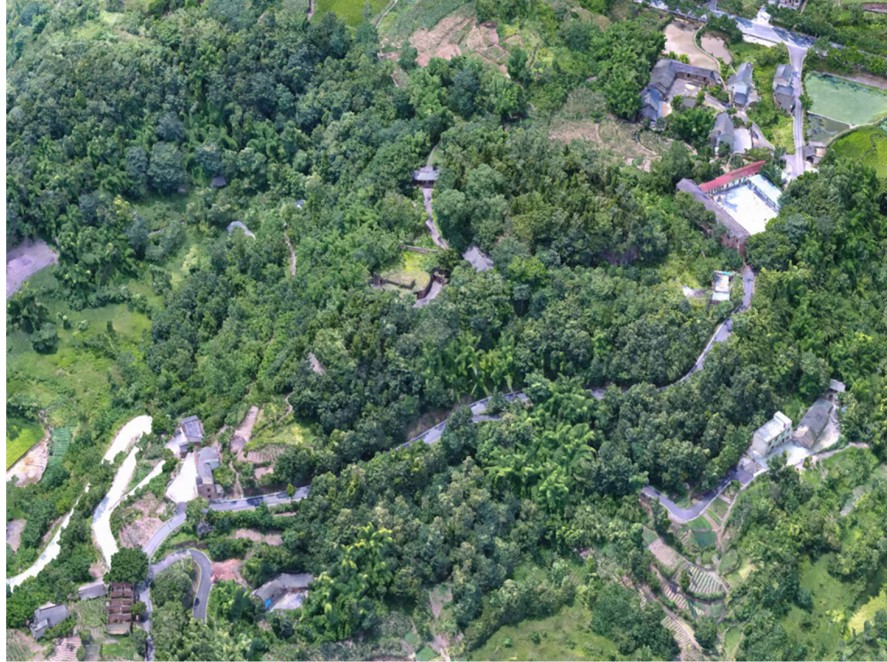

**Figure 5.** LOD0—The macro-scene in which the cultural relic is located was collected in 2015. A visualized 3D model was created by Beijing Digsur Science and Technology Co., Ltd. (Beijing, China) using the oblique photogrammetry technique with UAV data. Data was collected by UAVs, and dense point clouds were generated by oblique photography. Triangular mesh models were then constructed with textures.

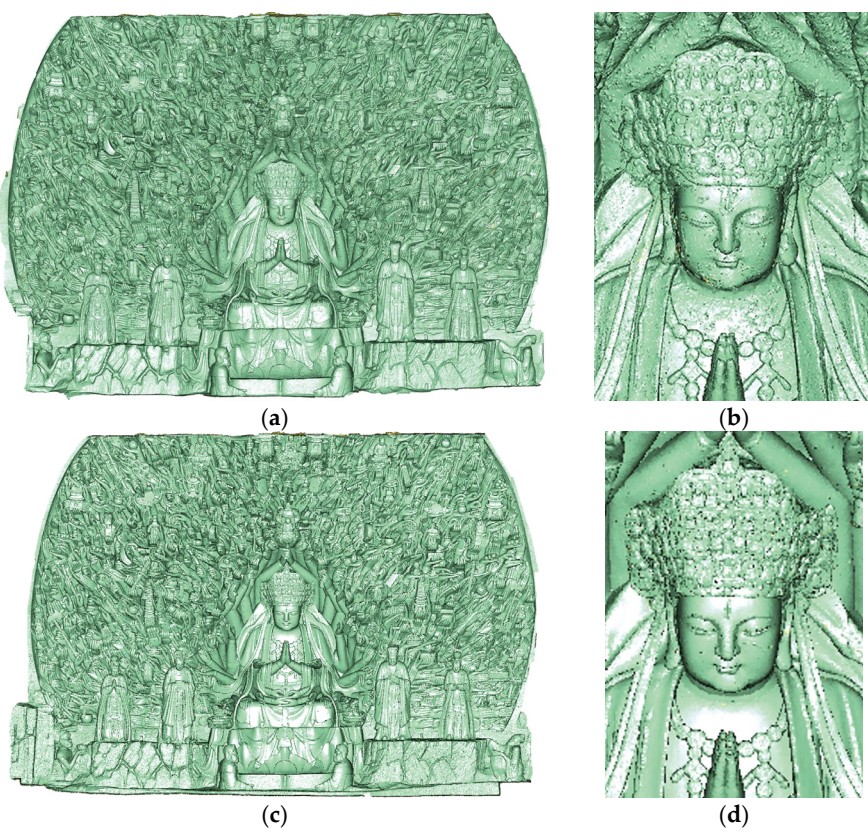

**Figure 6.** (**a**) LOD1—the meso-scene, comprising 3D point cloud data of the Dazu Thousand-Hand Bodhisattva Statue in 2011; (**b**) partial capture of point cloud data in 2011; (**c**) LOD1—3D point cloud data of the Dazu Thousand-Hand Bodhisattva Statue in 2015; (**d**) partial capture of point cloud data in 2015.

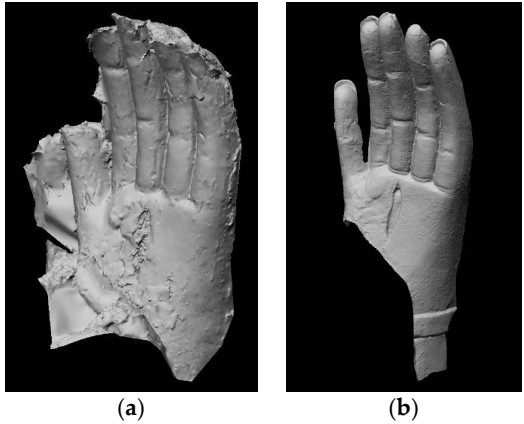

**Figure 7.** (**a**) LOD2—the micro-scene, comprising 3D point cloud of a Guanyin hand in 2011; (**b**) The 3D model is the result of virtual restoration, one special eye (heavenly eye) in the palm of hand and finger joints shown high ability to express geometric details.

Point cloud data does not contain semantic information. An information model is formed after the construction of the overall point cloud, or after a point set is associated with the semantic information. In this case, the information model contains two different time points. Each time point has three different spatial scales, and each point cloud model can be divided into point sets that are meaningful for cultural relic protection and associated semantics or files. This information model plays a vital role in the conservation and management of cultural heritage.

**Table 1.** Details of data collection.

| Year | LODs | Technology | Equipment | Data Description |
|------|------|-----------|-----------|-----------------|
| 2011 | LOD0 | - | - | - |
| | LOD1 | 3D laser scanning | Faro LS420 | Sample spacing 3 mm Sampling density 1 mm |
| | LOD2 | 3D laser scanning | CIM CORE Infinite 2.0 | Sample spacing 0.1 mm Sampling density 0.045 mm |
| 2015 | LOD0 | UAV tilt photography | FEIMA F200(UAV) Sony A7RII full-frame camera | Ground resolution 2.6 cm |
| | LOD1 | 3D laser scanning | Faro LS420 | Sample spacing 3 mm Sampling density 1 mm |
| | LOD2 | 3D laser scanning | CIM CORE Infinite 2.0 | Sample spacing 0.1 mm Sampling density 0.045 mm |

### 4.2.1. Investigating the Spatial Distribution of Surface Material at LOD1

Because the surface of the cultural relic reveals 3D complex geometric shapes, it is impossible to calculate the damaged area using an orthoimage. As shown in Figure 8a, a 3D mesh was constructed using the 3D point cloud data of the Dazu Thousand-Hand Bodhisattva Statue. The whole 3D point dataset was not associated with any attribute information. As a result, its application was limited to 3D visualization and basic geometric measurements, such as distance, volume, and area using the spatial information of the point clouds. Subsequently, by manually selecting the area covered with gold leaf, semantic information was added to the point set in the covered area, such as (X, Y, Z, gold leaf-covered). Therefore, the gold foil covering of cultural relics was able to be generated to realize the management of the relics' surface material (see Figure 8b). In the same manner, semantic information concerning the occurrence of gold leaf damaged was added to the point set, as shown in Figure 8c. The 3D mesh with added semantic information enabled multiple visualizations based on the different attributes, and supports the computation of damaged areas. Finally, the surface area of the whole statue was calculated as 211.534 m$^2$, and the surface area of the gold leaf damage was 130.629 m$^2$. The area of damaged gold leaf accounted for 61.75% of the total area of the statue.

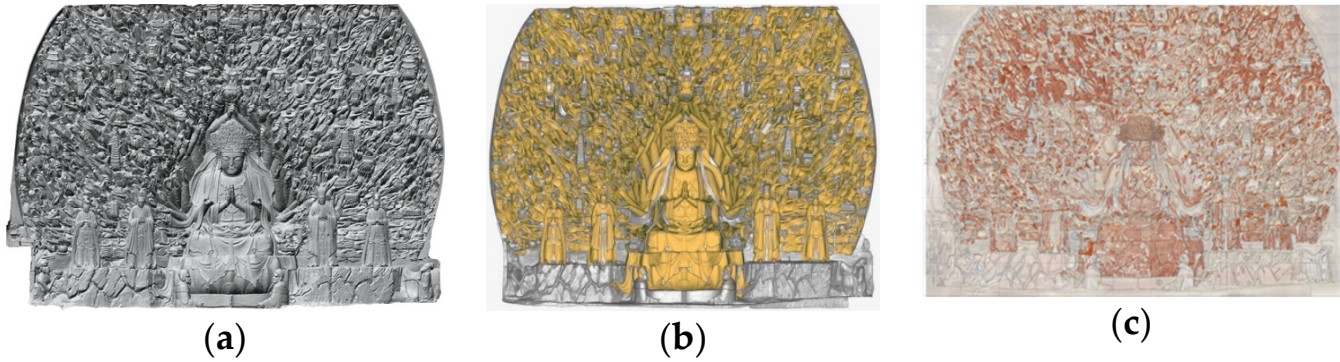

**Figure 8.** (**a**) LOD2—3D mesh of the Dazu Thousand-Hand Bodhisattva Statue in 2011; (**b**) the gilded area in yellow in 2011; (**c**) the damaged area of gold leaf in red.

### 4.2.2. Identifying Damaged Area at LOD2

Regardless of whether the damaged areas are automatically extracted from the point cloud or manually labeled, the distribution areas of different kinds of damaged parts are essential information for protecting cultural relics [64,65]. In the traditional approach, the damaged area of cultural relics is labeled using traditional 2D images. However, due to the occlusion of geometric structures, this approach makes it difficult to express the 3D shape

and details of cultural relics with complex geometric structures. In addition, the calculation of the damaged area using 2D images is not accurate. Figure 9 presents an example of a damaged information map, in which the points located at different damaged regions are associated with damaged types, as shown with different colors. Specifically, red indicates broken parts, green indicates the area of flaking paint, yellow indicates the area of gold leaf warping, and blue indicates the area where gold foil is missing.

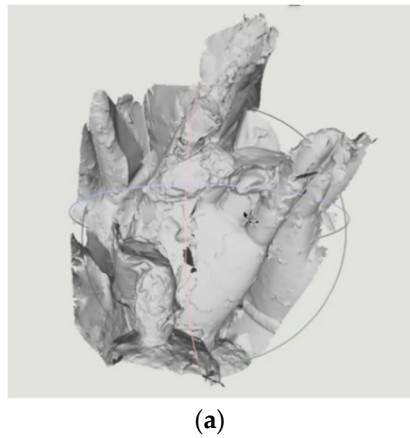 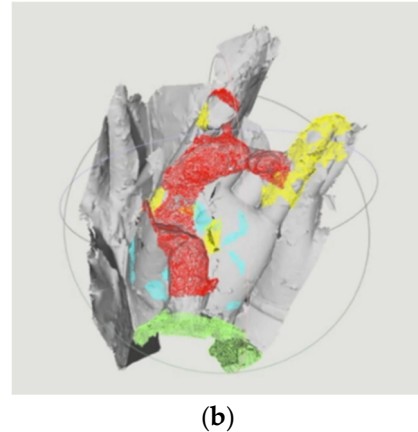

(**a**)                                   (**b**)

**Figure 9.** (**a**) 3D mesh of a hand from the Dazu Thousand-Hand Bodhisattva Statue; (**b**) different damage types are visualized in different colors.

### 4.2.3. Geometric Deformation Analysis at LOD2

Obtaining point cloud data for cultural relics at specific time intervals is important because it enables geometric deformation to be detected. The cause of geometric deformation is a major concern among cultural relic researchers. The smart point cloud modeling framework can correlate data collected at multiple timestamps to calculate the geometric changes in the artifacts through spatial analysis. Moreover, the model can continuously enrich semantic information via spatial analysis algorithms.

Figure 10a shows the mesh model of a damaged Guanyin hand. Figure 10b shows the virtual restoration result (i.e., the restoration of the damaged parts using a computer) based on the mesh model of the damaged hand. For this purpose, we used the grid editing function provided by the Geomagic Studio commercial software (https://www.3dsystems.com/ accessed on 25 July 2021). Furthermore, the actual repair work was carried out by referring to the virtual restoration result model shown in Figure 9b. The Hausdorff distance was used to calculate the degree of variation by comparing the two models in Figure 10a,b, which were at the same spatial scale [66]. Different colors represent the degree of variation and add additional semantic information to the point cloud. The red area indicates the variation degree between the two models is less than 20%, the yellow area indicates the variation degree is between 20% and 40%, the green area indicates the variation degree is between 40% and 60%, and the light blue area indicates the variation degree is more than 60%. Although successful attempts were undertaken, difficulties in the overlap of these two models remain. For example, the accuracy of the matching of corresponding points relies mainly on the selection of these points. Different people may select different corresponding points, resulting in randomness and uncertainty, which affects the final result of geometric deformation analysis, e.g., the quantitative variation degree.

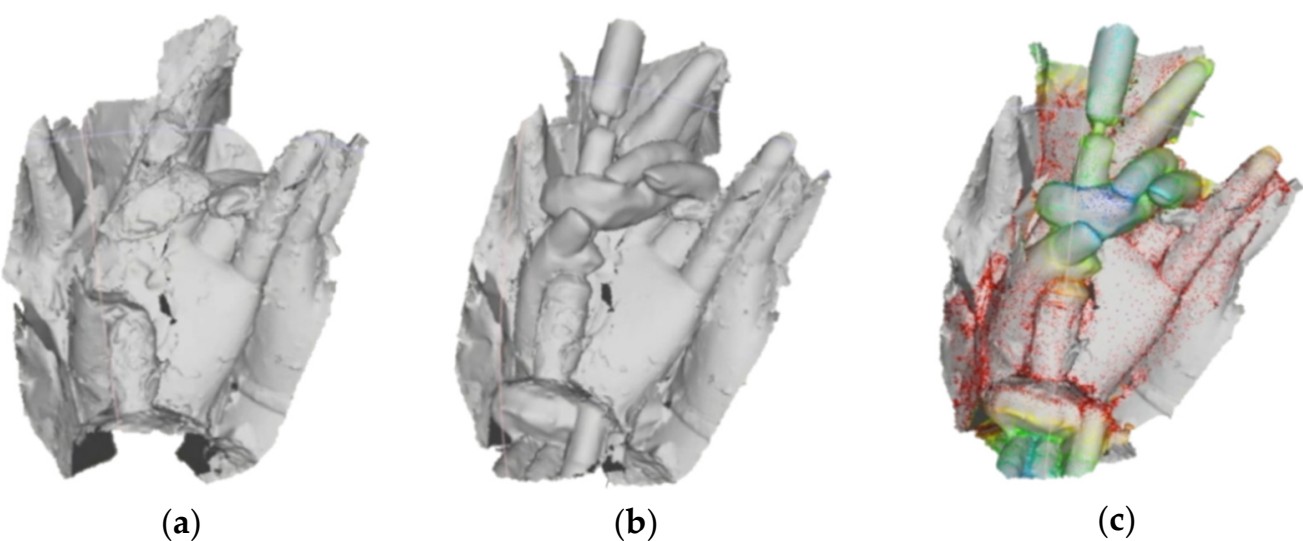

|       |       |       |
|:-----:|:-----:|:-----:|
| (**a**) | (**b**) | (**c**) |

**Figure 10.** (**a**) Three-dimensional mesh model of the damaged status; (**b**) geometric shape after virtual restoration; (**c**) comparison results of before and after restoration; the colors from red to blue represent the degree of geometric change.

## 5. Discussion

In a practical application of a smart point cloud of cultural relics with complex geometries, point cloud models of different spatial resolutions of the Dazu Thousand-Hand Bodhisattva Statue from the macro-scene (LOD0), the cultural relic itself (LOD1), and the cultural relic's details (LOD2), were established. These data were linked based on spatial scales. Thereafter, point clouds or subsets of point clouds were semantically related to the material and damage information of the relic. The point cloud contains attributes; thus, it can be used to visualize the spatial distribution of the cultural relic, to support the investigation of damaged areas and engineering drawings. In addition, because the data of two time periods can be obtained, the geometric changes in a cultural relic over time can be obtained through change detection. This fully demonstrates the potential of applying smart point clouds for relics having complex geometries.

The main difference between the modeling method proposed in this paper and HBIM is based on the point cloud model. HBIM is a parametric modeling method for geometric components, and one of its purposes is to reduce the size of the model. The second difference is the parameterized expression form, which can be reused for recurring components, thus reducing the time cost of information modeling. However, the disadvantage of HBIM modeling is that, using this approach, it is difficult to accurately express geometric textures in the parameterization of complex geometric shapes. Additionally, HBIM is mainly limited to historic architecture, and cannot by easily adapted to other types of cultural relics, such as statues of various sizes. The method proposed in this article can be applied to a broader range of relics and scenarios. When using our method for modeling, it should be noted that:

(1) Information related to cultural relics is hugely fragmented and involves multiple disciplines. Some materials are old and have a wide variety of information. Therefore, it is crucial to organize the data when constructing a model.

(2) When investigating the historical background and essential details of cultural relic objects, it is also necessary to pay attention to the hierarchical perspective during the interpretation of the cultural relic information. The historical background of cultural relic information is also an essential reference.

(3) When segmenting and clustering point cloud data, attention should be paid to the geometric structure, original form, and historical function to ensure that the results of this process can meet human cognition requirements.

In the digitization of cultural heritage, deep learning technologies [67–69] have been widely used for semantic segmentation of 3D point cloud datasets. These advanced data

processing technologies can improve the efficiency of modeling and reduce time and labor costs. However, engineering drawing production, feature extraction, classification, and segmentation still require significant manual intervention in real-world applications.

## 6. Conclusions

In this study, smart point clouds of cultural relics with complex geometries were proven to be an information model framework with potential in managing, protecting, and applying cultural heritage. This approach emphasizes the correlation between point cloud data in different special scales and time periods, in addition to the correlation between point clouds and semantics or files. At present, in the field of cultural heritage, independent data cannot support the research of multiple disciplines and different stakeholders. Thus, information integration plays a significant role in data sharing. Due to the differences among cultural relics, this paper does not present detailed proposals of either key algorithms or implementation technologies, but emphasizes the framework of smart point clouds and the benefits of this approach for cultural heritage. Smart point clouds of cultural relics result from the logical integration of multisource data, and contain a large amount of semantic information. This provides a framework for the use of digital cultural relic data to realize greater potential. Semantic segmentation of point clouds via machine learning methods has the potential to make the framework more automatic and intelligent, which is an important future research direction.

A case study showed that the proposed method is feasible. This framework balances the contradiction between geometric accuracy and semantic information of 3D models, particularly for artifacts with complex geometric structures, and has broad applicability and strong expansion potential. This information modeling framework can be used to guide the design of conceptual and logical models in the cultural heritage information system. This method allows flexible queries and retrieval between datasets (e.g., point cloud data and point cloud data, point cloud data and their subsets, point cloud data and semantics, point clouds and other files). Due to the integration of semantic information with spatial information, spatial analysis can be carried out at different scales, such as 3D GIS. The spatial distribution of additional attributes provides a reference for thematic engineering drawing.

**Author Contributions:** Conceptualization, Su Yang and Miaole Hou; methodology, Su Yang and Miaole Hou; validation, Su Yang; formal analysis, Su Yang; investigation, Su Yang, Miaole Hou, Songnian Li and Ahmed Shaker; writing—original draft preparation, Su Yang; writing—review and editing, Su Yang, Songnian Li and Ahmed Shaker; visualization, Su Yang; supervision, Songnian Li; project administration, Miaole Hou; funding acquisition, Miaole Hou. All authors have read and agreed to the published version of the manuscript.

**Funding:** This research was funded by Beijing Natural Science Foundation, grant number KZ202110010 6021. National Natural Science Foundation of China, grant number 4217012259.

**Institutional Review Board Statement:** Not applicable.

**Informed Consent Statement:** Not applicable.

**Conflicts of Interest:** The authors declare no conflict of interest.

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
