# Peer review of "Modeling and Processing of Smart Point Clouds of Cultural Relics with Complex Geometries"

_ijgi, doi:10.3390/ijgi10090617_

Round 1
Reviewer 1 Report
I would suggest extending the experimental workflow by adding more technical information on segmentation processes. Is the segmentation manual or did you use some ML algorithms based on shape analysis? This is a crucial aspect to make the whole process repeatable even on even more complex case studies or even just to set the limits of this application to objects with specific features.
Reviewer 2 Report
This paper proposed a modelling framework for cultural relics with complex geometries based on a 3D point cloud. In general, the article is well structured and well-illustrated. The study is interesting for the management and monitoring of cultural heritage. However, some issues in the paper should be better explained or improved.
Major comments:
- The type of sensors (active or passive) used for the 3D acquisition of point cloud should be exposed in the subsection “point cloud” to better understand the nature of these data. For example, the 3D point cloud acquisition from laser technology and UAV is different.
- The explanation of the modelling framework is well-written. However, more description about the methods is needed. For example: What’s the technique used to classify the 3D point cloud (clustering, segmentation)? Is it have tasks of the framework that are automatic? How is the semantic information linked to the point cloud? How are the damage areas identified on the object?
- The object chosen to test the performance of the modelling framework was a good choice, taking into account the complexity of the object and was restored for a period. However, some details in section 4 should be improved, such as more information about the point cloud used and the type of UAV used ( multi-rotor or fixed-wing, or other).
- Please, review the use of “modeling” or “modelling” in the paper.
Specific Comments:
Ln88: What’s the meaning of high-resolution point clouds?
Ln221: The 3.3.1 subsection about “Extracting salient features” doesn’t appear in the scheme of figure 1. Explain better the “extracting salient features” method.
Ln177: Why is it difficult to model the complex structure? Improve this sentence.
Ln193:The legend should be improved; Does the time in scheme equal to 1…1?
Ln195: “Geometric deformation is related to ..the geometric structure..”. Clarify this sentence.
Ln285: The framework represented in figure 4 seems to focus only on the specific elements of the statue (like hands) and damaged areas. For example, where is the identification of areas not damaged classified in LOD1 and LOD2?
Ln304: “In addition, the area of the disease can be calculated.” Can we explain better how the area is calculated? Directly from the point cloud? After applying the segmentation or clustering?
Ln317: Table 1, What’s the meaning of accuracy for the last column? What’s the meaning of position exposed here? How does the sampling density represent accuracy? Please, clarify better this information.
Ln320: Does the LOD0 showed in figure 1 an orthoimage/orthomosaic obtained from the processing of UAV imagery? Please, clarify.
Ln332-339. Please, rewrite this paragraph. In addition, what’s the percentage of golf leaf area damaged in the statue?
Ln 356-362: Geometric deformation analysis at LOD2 should be better described. For example, what’s the method used for the “overlapping” of the two 3D models? Do the 3D models have the same scale? What are the difficulties found in the overlap of these two models?
Some additional comments are given in the pdf document.

Reviewer 3 Report
The document is not very original, but it is a good practice on "how to write a single scientific article". The case study is described in a very clear and precise manner, the methodology used is correct, the conclusions - maybe too obvious - are always correct. however, this is a highly informative text, lacking technical information, tables and graphs that may allow us to judge the quality of the results. The paper should be written more as an engineer than as a scientific writer.
